# More than Meets the Eye: Using Textural Analysis and Artificial Intelligence as Decision Support Tools in Prostate Cancer Diagnosis—A Systematic Review

**DOI:** 10.3390/jpm12060983

**Published:** 2022-06-16

**Authors:** Teodora Telecan, Iulia Andras, Nicolae Crisan, Lorin Giurgiu, Emanuel Darius Căta, Cosmin Caraiani, Andrei Lebovici, Bianca Boca, Zoltan Balint, Laura Diosan, Monica Lupsor-Platon

**Affiliations:** 1Department of Urology, “Iuliu Hatieganu” University of Medicine and Pharmacy, 400012 Cluj-Napoca, Romania; t.telecan@gmail.com (T.T.); drnicolaecrisan@gmail.com (N.C.); emanuelcata@yahoo.com (E.D.C.); 2Department of Urology, Clinical Municipal Hospital, 400139 Cluj-Napoca, Romania; lorin.giurgiu@gmail.com; 3Department of Medical Imaging, “Iuliu Hatieganu” University of Medicine and Pharmacy, 400012 Cluj-Napoca, Romania; cosmin.caraiani@umfcluj.ro (C.C.); andrei1079@yahoo.com (A.L.); bianca.petresc@gmail.com (B.B.); monica.lupsor@umfcluj.ro (M.L.-P.); 4Department of Radiology, Emergency Clinical Country Hospital, 400006 Cluj-Napoca, Romania; 5Department of Radiology, “George Emil Palade” University of Medicine, Pharmacy, Science and Technology, 500139 Târgu Mureș, Romania; 6Department of Biomedical Physics, Faculty of Physics, “Babes-Bolyai” University, 400084 Cluj-Napoca, Romania; zoltan.balint@ubbcluj.ro; 7Department of Computer Science, Faculty of Mathematics and Computer Science, “Babes-Bolyai” University, 400157 Cluj-Napoca, Romania; laura.diosan@ubbcluj.ro; 8Department of Medical Imaging, Regional Institute of Gastroenterology and Hepatology “Prof. Dr. Octavian Fodor”, 400162 Cluj-Napoca, Romania

**Keywords:** prostate cancer, multiparametric magnetic resonance imaging, textural analysis, artificial intelligence, radiomics, computer-assisted diagnosis

## Abstract

(1) Introduction: Multiparametric magnetic resonance imaging (mpMRI) is the main imagistic tool employed to assess patients suspected of harboring prostate cancer (PCa), setting the indication for targeted prostate biopsy. However, both mpMRI and targeted prostate biopsy are operator dependent. The past decade has been marked by the emerging domain of radiomics and artificial intelligence (AI), with extended application in medical diagnosis and treatment processes. (2) Aim: To present the current state of the art regarding decision support tools based on texture analysis and AI for the prediction of aggressiveness and biopsy assistance. (3) Materials and Methods: We performed literature research using PubMed MeSH, Scopus and WoS (Web of Science) databases and screened the retrieved papers using PRISMA principles. Articles that addressed PCa diagnosis and staging assisted by texture analysis and AI algorithms were included. (4) Results: 359 papers were retrieved using the keywords “prostate cancer”, “MRI”, “radiomics”, “textural analysis”, “artificial intelligence”, “computer assisted diagnosis”, out of which 35 were included in the final review. In total, 24 articles were presenting PCa diagnosis and prediction of aggressiveness, 7 addressed extracapsular extension assessment and 4 tackled computer-assisted targeted prostate biopsies. (5) Conclusions: The fusion of radiomics and AI has the potential of becoming an everyday tool in the process of diagnosis and staging of the prostate malignancies.

## 1. Introduction

Prostate cancer (PCa) is the most diagnosed urological malignancy in the male population [1]. Any clinical or biochemical suspicion of neoplasia requires further investigations via multiparametric magnetic resonance imaging (mpMRI) [2], each lesion being classified according to Prostate Imaging-Reporting and Data System (PI-RADS) scoring [3]. However, although the mentioned scale has the property of raising the suspicion of PCa with high positive predictive value, according to current guidelines, a histopathological confirmation of PCa and aggressiveness evaluation is needed in order to decide upon further therapeutic strategies, thus more invasive procedures are being imposed.

Targeted prostate biopsy is performed based on radiologist’s interpretation of mpMRI acquisitions. Over the years, significant inter-observer variability has been reported in terms of PI-RADS score assessment, as the overall agreement between radiologists reaches only 61% [4], thus not providing a fully reliable decision tool to stratify the indication of prostate biopsy. Moreover, when it comes to MRI-guided targeted sampling, the manual annotation of described nodules and synchronization between T2 sequences and real-time transrectal ultrasound images, in most departments, it is carried out by a urologist. Therefore, the risk of missing clinically significant PCa (csPCa) found outside of the selected region of interest is present in 27.6% of cases [5]. Additionally, mpMRI plays a crucial role in PCa staging. Even so, the sensitivity reported by the multicenter study conducted by Kam et al. [6] reaches only 38%, showing that additional factors that have the potential to weight in upon the staging are needed.

As a response to these hindrances, Lambin et al. [7] introduced the concept of radiomics for the first time in 2012, being defined as a novel domain meant to extract information derived from medical imaging acquisitions, using AI-based and similar image processing techniques. Texture analysis represents a subdomain of radiomics that focuses on quantifying the heterogeneity of pixels in selected regions of interest, finding its applicability in oncological imaging. Recent studies highlighted the possibility of implementing radiomics features into routine PCa diagnostic workflow. Nketiah et al. [8] showed an increased csPCa detection rate when texture analysis was implemented as compared to standard targeted prostate biopsy protocol (84% versus 56%). The combination of automatic detection of suspect prostate nodules and textural features characterization has the potential to become an AI-based “prostate biopsy”, that will discern benign from aggressive cancerous tissue, thus becoming a non-invasive diagnostic solution [9].

This paper aims to present the current state of the art regarding decision support tools based on texture analysis and artificial intelligence for MRI image analysis and to assess their accuracy and performance in terms of PCa diagnosis and staging, prediction of aggressiveness as well as biopsy assistance.

## 2. Materials and Methods

A thorough PubMed MeSH, Scopus and WoS (Web of Science) search was performed in February of 2022, targeting the topic of multiparametric magnetic resonance imaging (mpMRI) and decision support systems in diagnosing and staging of prostate cancer. Keywords included (but not limited to) the following: “prostate cancer”, “MRI”, “radiomics”, “textural analysis”, “artificial intelligence”, “computer assisted diagnosis”. Retrieved articles were selected or excluded based on specific criteria:

Inclusion criteria:Only original research papers were considered;Decision support tools trained and validated on at least 50 cases;Imagistic technique employed: mpMRI, with specified field strength (1.5 or 3 T);Analytical observational studies, written in English and published in the last 10 years;Focus on clinical aspects.Exclusion criteria:Study population under 50 cases;Other imagistic methods used, including biparametric MRI (bpMRI);Papers designed as systematic reviews, meta-analyses, comments, letters to editor, case reports and clinical practice guidelines;Articles focusing on the technical aspects of MRI, textural analysis and artificial intelligence, without a well-established clinical application;Studies based on public datasets or carried out on animal subjects or phantom substitutes.

Additional studies were included by reviewing the reference lists of the selected studies, that were missed through the aforementioned search strategy. The search process was synthetized in a Preferred Reporting Items for Systematic Reviews and Meta-analyses (PRISMA) type flowchart (Figure 1).

## 3. Results

The obtained results were categorized according to the three main criteria as stated in Figure 1. Diagnostic accuracy and prediction of PCa aggressiveness (24 studies), diagnostic accuracy and prediction of extracapsular extension (7 studies) and artificial intelligence-assisted targeted prostate biopsy (4 studies).

### 3.1. Diagnostic Accuracy and Prediction of PCa Aggressiveness

#### 3.1.1. General Data

Following the above-mentioned inclusion criteria, 24 studies published between 2017–2021 have been selected (Table 1). A multicenter research protocol has been described in 20.83% (*n* = 5) of papers. The median number of cases per study was 222, ranging from a minimum of 54 subjects, to a maximum of 1034. In nearly half of the studies (*n* = 11), the subjects were divided into a ‘training’ and a ‘testing’ cohort, 25% (*n* = 6) of the described protocols had an additional ‘validation’ group, while 29.16% (*n* = 7) did not elaborate the division of patients. All studies were conducted using mpMRI scans, 66.6% (*n* = 16) reporting a field power of 3 Tesla (T), 20.83% (*n* = 5) 1.5 T and 12.5% (*n* = 3) used 2 different MRI scanners of 1.5 T and 3 T, respectively. Texture analysis features were extracted from more than one acquisition in 75% of papers (*n* = 18), T2-weighted images being the most constantly used (83.33%, *n* = 20). The segmentation (identifying the region of interest on each mpMRI slice) was done in an automated or semiautomated fashion in 16.66% (*n* = 4) of cases. The ground truth used as reference by texture analysis software was the pathological result from prostate biopsies in 58.33% (*n* = 14) of studies, from radical prostatectomy specimens in 25% (*n* = 6), or from both in 16.66% of the reported studies (*n* = 4).

#### 3.1.2. AI-Based Automatic Detection of PCa

Khosravi et al. [23] developed an automated prediction model, built on deep learning algorithms, that extracted relevant features from T2 weighted images, using class activation maps (AI based algorithms that emulate neuronal networks, used to discriminate between a given set of images) and GoogLeNet architecture. The system was trained on 212 cases, 95 of them having biopsy confirmed PCa with Gleason Group (GG) 3–5, while 117 were classified with benign lesions. The algorithm was constructed to discern non-malignant nodules from csPCa, reaching an accuracy of 81.8%.

Another paper, published by Giannini et al. [14], focused on creating a computer-assisted diagnosis tool, based upon texture analysis features derived from both T2WI and ADC maps, that aimed to automatically differentiate cancerous nodules from indolent ones. The system was trained on 81 lesions, confirmed after radical prostatectomy, and validated on two groups, with 38 and 30 cases each, from two different centers. While the accuracy was as high as 95.1% in the training setting, it dropped significantly to 75% in the validation groups, due to the data originating from two different MRI machines.

To achieve higher performance, various authors tried to associate multiple acquisitions, or to subtract volumetric parameters. The highest performance was achieved when DCE features were associated with non-contrast-enhanced sequences, providing additional input information for machine learning models and reaching higher sensitivity, specificity and AUC compared to gradient-boosting machines based solely on T2WI and DWI (100% versus 86%, 100% versus 90%, 1 versus 0.953, *p* = 0.001). This shows that the tumor’s microvasculature perfusion is an earlier indicator than the diffusion of water molecules on DWI and ADC maps [16]. A parameter that could potentially characterize malignant prostatic tissue was tumor shape, Cuocolo et al. [26] describing a significant link between ADC-derived surface area to volume ratio (which measures the degree of tumor compactness and spheroid shape) and csPCa, with a sensitivity, specificity and AUC of 56%, 97% and 0.78, respectively (*p* = 0.002).

#### 3.1.3. Prostate Cancer Aggressiveness

The greatest foreseeable advantage that radiomic features might bring is the possibility of differentiating between indolent and clinically significant PCa, providing a non-invasive way to conduct active surveillance strategies, limiting the use of prostate biopsies, or deciding the treatment strategy, with better performance than PI-RADS classification (sensitivity, specificity and AUC of 90% versus 83%, 70% versus 47% and 0.85 versus 0.73, respectively, all *p* < 0.05) [30].

Aiming to develop an ‘AI-biopsy’, Min et al. [28] analyzed the radiomic signature of csPCa in 280 patients confirmed with malignant nodules via MRI-TRUS fusion prostate biopsy. The authors reported a significant difference between clinically insignificant and csPCa (*p* < 0.01), concluding that texture analysis has the potential of further stratifying the indication of biopsy.

The study conducted by Chen et al. [24] elaborated a texture analysis-based protocol, that aimed to differentiate PCa from non-PCa lesions, as well as low-risk PCa from high-risk PCa. Having a larger sample population than the previous studies, the authors showed a significantly higher AUC in terms of differentiating cancerous from non-cancerous lesions for the prediction model based on T2WI and ADC maps, compared to PI-RADS score (0.99 versus 0.86). The same results have been reported for comparing low-risk PCa versus high-risk PCa; however, due to the limited number of low-grade tumors, the synthetic minority oversampling technique (SMOTE) was applied, a technique that amplifies the number of available relevant cases in a small cohort, in order to obtain a statistically significant result.

Finally, texture analysis could further stratify PI-RADS 3 lesions into indolent or clinically significant PCa (csPCa) nodules, thus avoiding unnecessary biopsies [12,19,21]. The paper published by Hectors et al. [12] showed a sensitivity and specificity of 75% and 79.6%, respectively, with an AUC of 0.76 regarding the detection of csPCa among PI-RADS 3 lesions (*p* = 0.022), while mpMRI alone registered a sensitivity of 38.24% and a specificity of 53.85% for PI-RADS 3 tumors [34].

#### 3.1.4. Decision Support Tools’ Accuracy Compared to Radiologists’ Interpretation

Out of the 24 papers analyzed, 3 compared the performance of decision support systems with the results given by senior radiologists, with expertise varying from 4 to 10 years [11,22,31]. The study conducted by Bonekamp et al. [11] reported a significantly higher diagnostic performance than the radiologists for features extracted from ADC maps, for both training and testing groups (*p* = 0.008 and 0.048, respectively), while the radiomic machine learning model outperformed the radiologist only in the test setting. Castillo et al. [22] developed a predictive model based on textural features extracted from T2WI, ADC maps and DWI acquisitions, with higher diagnostic accuracy as compared to two senior radiologists (average AUC = 0.75 versus 0.5 and 0.44 for the two experts involved), but only when the training and testing subjects came from the same center. For the external validation cohort, mean AUC dropped to 0.54, most likely due to the dependency of radiomic features on MR acquisition setup and manual delineation being made by different radiologists, based on pathology reports made by different specialists. In terms of attributing a PI-RADS score to a suspect lesion, the prediction of the artificial intelligence model proposed by Sanford et al. [31] overlapped in 58% of cases with the radiologist’s result. Interobserver agreement between the radiologists and the AI-based model increased proportionally with the PI-RADS score: from 6% for PI-RADS 2 lesions to 80% for PI-RADS 5 cases. Overall, there was no significant difference between the deep-learning system and radiologist’s prediction of aggressiveness (*p* = 0.59, 0.36 and 0.47 for lesions of PI-RADS 3, 4 and 5, respectively). Although the results did not reach statistical significance, adding artificial intelligence elements to the diagnostic workflow might reduce the inherent subjectivity of assessing the PI-RADS score, thus limiting interobserver variability.

### 3.2. Diagnostic Accuracy and Prediction of Extracapsular Extension (ECE)

#### 3.2.1. General Data

We identified seven studies published between 2019–2021 (Table 2). Regarding the number of centers, 42.85% (*n* = 3) were designed as a multicentric study. The total number of cases was 2209, ranging from a minimum of 95 patients to a maximum of 849 patients. Following the study protocol, 85.71% (*n* = 6) of papers divided their subjects into a “training” and a “testing” cohort, while only one had an additional “validation” group. All studies were conducted using mpMRI scans, 85.71% of them employed a 3 Tesla power field mpMRI machine (*n* = 6) and 14.28% (*n* = 1) reported a multivendor strategy (both 1.5 T and 3 T). The most constantly used sequence was T2-weighted imaging (T2WI). For feature extraction, apparent diffusion coefficient (ADC) was used in 71.42% of the studies (*n* = 5) and diffusion-weighted imaging (DWI) with dynamic contrast-enhanced method (DCE) was used in 28.57% of the studies (*n* = 2). Regarding image segmentation, they were manually done in 85.71% of the cases (*n* = 6) and automated in only one case. The radical prostatectomy pathological report was the ground truth for all study protocols and least absolute shrinkage and selection operator (LASSO) regression algorithm was employed to build the radiomics model, to regularize the data used for training purposes, thus avoiding redundant variability and enhancing accuracy.

#### 3.2.2. AI-Based Tools for Automatic Detection of ECE

Most studies relied on manual segmentation of the prostate and surrounding structures. Hou et al. [35] conducted a multicentric study of 849 patients, developing a deep learning network that automatically detected ECE serving as a non-invasive, preoperatory tool, advising upon the oncological safety of nerve-sparing procedures. The algorithm performed better when only the slice with the maximum tumoral diameter was taken into consideration, compared to multislice-based prediction (AUC of 0.818 versus 0.799, *p* = 0.019). This finding was attributed to the hypothesis of overprediction in multislice analysis, leading to overstaging of PCa. When tested on external validation datasets, the accuracy dropped from 83.6% in the training phase to 71.8%, showing that, although based on many cases, the algorithm built on a single MRI machine and acquisition protocol is not perfectly reproductible when employed in a different setting. Similar results were published by Cuocolo et al. [36], in a study designed with two external validation cohorts, achieving an accuracy of 79% and 74%, respectively.

#### 3.2.3. Radiomic and Texture Analysis-Based Prediction of ECE

Several studies focused upon extracting significant textural features, to objectify the extraprostatic effraction [37,38,39,41]. For the selected papers, the sensitivity, specificity and accuracy of predictions reached 94.6%, 89.4% and 85.8% in the training setting and 84.6%, 72.7% and 81.8% when it was applied to an external validation cohort.

Bai et al. [37] included 284 patients with PCa from two centers, analyzing the radiomic signature of both intra- and peritumoral regions. Defined as the 3–12 mm surrounding area of the suspected nodule, the peritumoral region turned out as a better predictor of capsular effraction, motivated by similar changes of vessels and soft tissue found in the periprostatic area. Since the algorithm was trained on acquisitions obtained from 3 different MRI scanners, the performance maintained constant in the external validation setting as well.

Taking a step forward, foreseeing the ECE could eventually decrease the rate of positive surgical margins (PSM). A study conducted by He et al. [38] assessed radiomic features that were strongly correlated with PSM on the radical prostatectomy specimens. The highest accuracy, of 72.8%, was achieved by ADC-extracted parameters, thus correlating with the tumor’s cellularity and cell count.

Although the results seem to be promising as stand-alone variables, the final target of radiomic signatures is to provide another puzzle piece to the diagnostic workflow and to stratify the indication of invasive procedures and radical treatments. Combining the clinicopathological data such as age, total and free PSA, PI-RADS score and Gleason group with T2, ADC and DWI-derived radiomic features, the accuracy of extracapsular extension prediction increased from 67.1% to 85.4% in the training group and from 69.7% to 81.8% in the validation setting [38].

#### 3.2.4. The Accuracy of the Decision Support Tool Compared to the Interpretation of the Radiologist

Out of the analyzed seven papers, three aimed to compare the preoperative probability of ECE given by radiomic assessment with the radiologists’ prediction [35,36,40]. All studies showed that texture analysis features were superior compared to best radiologists’ performance, reaching an accuracy of 84.62% versus 73.43% (*p* = 0.02) and a sensitivity of 76.81% versus 60.87% (*p* = 0.043). Regarding the prostate segment, the radiomics model showed the best prediction capacity when assessed the apex of the prostate, due to the anatomical constrains of the region that hinder the radiological diagnosis, such as the absence of distinct capsule contour and neighboring structures such as the neurovascular bundles and external sphincter [40]. When the experts used artificial intelligence and radiomics model to adjust their ECE diagnosis, the accuracy increased from 67.6% to 79.5% in the training setting and from 64.7% to 76% in the validation cohort, concluding that deep learning prediction models have the potential of serving as a daily decision support tool [35].

### 3.3. Artificial Intelligence-Assisted Targeted Prostate Biopsy

#### 3.3.1. General Data

After screening the literature, four papers met the inclusion criteria (Table 3). The number of patients per study ranged between 62 and 916, being selected from multiple centers in 75% of cases (*n* = 3). The research protocol was heterogenous, with one study comprising a “training” and a “testing” cohort, while the others did not detail the validation process of the artificial intelligence-driven prediction algorithm. Most of the studies aimed to use a single vendor mpMRI scanner of 3 Tesla magnetic power field (*n* = 3), and one used 3 different machines of 1.5 and 3 Tesla, to increase the training data variability degree. The most frequently used acquisition was T2WI. The ground truth used as reference was the pathological report of the radical prostatectomy specimen in one of the studies and targeted biopsy cores in the other three studies.

#### 3.3.2. Accuracy and csPCa Detection Rate of AI-Assisted Targeted Prostate Biopsy

The first improvement brought by artificial intelligence was the automatic segmentation of the prostate prior to targeted biopsy. Soerensen et al. [42] developed a deep-learning model that performed the delineation of the gland 17 times faster than radiology technicians, while maintaining the same accuracy, thus sparing approximatively 16 work hours per 100 patients.

MRI-targeted biopsy is known to increase the csPCa detection rate, compared to standard systematic biopsy. However, it is still highly dependent on the mpMRI interpretation given by the radiologist, as well as the image registration performed by the urologist during the fusion process. In order to achieve a 95% detection rate of high Gleason grade tumors, van de Ven et al. [43] concluded that the required target accuracy for tumors of at least 0.5 cm [3] is 1.9 mm, such precision being achievable by employing AI-tools.

Regarding csPCA detection rate (CDR), Campa et al. [44] compared the performance of an automated computer-assisted prediction model with the analysis provided by senior uro-radiologists with 10 and 15 years of experience. Targeted cores were sampled from nodules described by the automated prediction system and confirmed by experts (‘’target-in-target” lesions, TiT), as well as from suspect areas identified solely by radiologists (RAD) or by the computer-assisted diagnosis algorithm (CAD). The cancer detection rate increased from 68.64% for nodules sampled based on RAD assessment to 81.81% for TiT cases. Moreover, 78% of the highest Gleason scores of the study cohort were detected by target-in-target biopsies, thus avoiding unjustified active surveillance strategies. A similar paper published by Ferriero et al. [45] studied the CDR with and without CAD assistance. The authors concluded that the AI system had the greatest advantage in nodules located in the anterior and transitional zone, increasing the detection rate from 11.1% to 54.5% (*p* = 0.028), CAD being the only independent prediction factor for csPCa detection in the above-mentioned regions (*p* = 0.013).

## 4. Discussion

We aimed to highlight the progress of artificial intelligence and its use in daily clinical practice, being a valuable tool for diagnosing and staging prostate cancer. Radiomics, and especially texture analysis, quantify the heterogeneity of selected regions of interest compared to the surrounding structures, having proven their utility in detecting and characterizing malignant tissue [46]. Based on extracted features, machine-learning algorithms can automatically identify suspect lesions, distinguishing indolent from csPCa and ultimately assessing preoperative extracapsular extension with an accuracy of 83.58% compared to the prostatectomy specimen [40], thus serving as a decision support tool regarding further treatment.

### 4.1. PCa Detection and Aggressiveness

Currently, the component with the highest precision of the PCa diagnostic workflow is mpMRI, reaching an accuracy of 60% and 83% for detecting csPCa in nodules graded as PI-RADS 4 and 5, respectively [47]. However, intermediate-risk patients with PI-RADS 3 lesions reach an accuracy as low as 27%, mainly explained by the poor interobserver agreement, estimated to be around 43% [4]. From this standpoint, assisted diagnosis has the potential of improving csPCa detection rates, especially when used by less-experienced radiologists. Hambrock et al. [48] demonstrated that readers with less than 50 prostate mpMRI interpretations improved their overall, PZ and TZ accuracy by 10% (*p* 0.001), 9% (*p* < 0.001) and 7% (*p* = 0.01), respectively.

Regarding the evidence provided by the selected papers, some limitations can be addressed. Firstly, although the total number of subjects respected the chosen inclusion criteria, when divided into testing and training cohorts, 9 studies allocated under 50 cases for testing their developed model, thus raising the question of accuracy of the reported results [10,13,14,15,20,25,26,27,29]. In terms of cohorts’ composition, 2 studies designed the training group on a multicentric structure, using different MRI machines or mixing in-house cases with publicly available ones [22,23], while other studies trained, tested and validated the algorithm on patients selected from the same center [10,30]. Although the authors suggest that exposing neural networks to various MRI settings reduces the risk of overfitting the algorithm to one center, CAD needs sufficient data variety in order to reach high diagnostic accuracy, one of the main sources being multivendor MRI acquisitions. However, the number of cases included in the selected papers might be insufficient and the desired variability will not be properly represented by a significant number of cases for decision support tools development. Second, studies focusing on aggressiveness prediction encounter limitations in terms of differentiating each Gleason score, mostly classifying each lesion as below or above a Gleason score of 7 [14,18,27,29,33]. The authors attributed this limitation to the insufficient number of cases required for a subdivision of patients based on Gleason score. In this context, patients cannot be further stratified into low-risk, intermediate-risk and high-risk PCa groups, thus implying that all csPCa lesions benefit from a unitary treatment. Lastly, a frequently admitted limitation is the lack of differentiation between peripheral and transitional zone nodules. While some studies purposefully limit the investigation protocol to the peripheric lesions, being considered more obvious and well-defined, thus suitable for designing a PCa detection prototype [10,14,16], others were restricted by the sample population size, which did not allow a separate analysis of peripheral and transitional lesions [12,24,25,26,29].

### 4.2. Extracapsular Extension Assessment

Preoperatory extraprostatic involvement is defined by specific digital rectal examination and imagistic findings. However, both methods have clear limitations, as DRE has an average accuracy below 60% [49], while the rate of upstaging to extracapsular extension at the final pathological report compared to the mpMRI prediction is up to 29% [50]. Using radiomics-based computer-assisted algorithms to weight in upon the diagnosis of ECE increased the radiologists’ accuracy up to 76% [35].

The main limitation that this subgroup of papers faced was their retrospective nature, having no possibility of investigating and adjusting radiomic features that affect the reproducibility of the prediction model [35,36,38,40,41]. The data heterogeneity is further accentuated by manual annotation of the suspect lesions, thus making the delineation process highly dependent of the radiologist’s experience and making the external validation rather difficult and unreliable [36,37,38,39,40,41].

Another identified hindrance is the employment of only T2 acquisition into the feature extraction process [40,41]. Although the anatomical criteria for extracapsular extension are mostly subtracted from T2 images, it is considered that functional sequences that provide diffusion details might increase the overall accuracy and reproducibility.

Lastly, the best performance of the prediction model in an AI-radiologist based setting was achieved in large tumors, with a higher PI-RADS score and D’Amico risk score, thus leaving the open invitation for future papers to assess ‘grey-zone’ lesions, with intermediate levels of PSA and a PI-RADS score of 3 [35].

### 4.3. AI-Assisted Targeted Prostate Biopsy

The main hindrance that targeted biopsy faces is the lack of guidance for the urologists when performing the lesion annotation. Although targeted sampling increases the diagnostic rate of csPCa from 48% to 56% when compared to standard systematic biopsy, adding textural analysis extracted features can increase the biopsy accuracy up to 84%, however only when used in combination with the radiologist’s interpretation [8].

The researched literature is relatively scarce, with little insight regarding computer-assisted targeted biopsy, most studies not meeting the inclusion criteria or focusing on other imagistic techniques than mpMRI. Moreover, each individual paper had a different study protocol, thus drawing a general conclusion is rather difficult, with a level of evidence of low strength.

Soerensen et al. [42] developed an algorithm that managed to automatically contour the prostate outline. The main identifiable hindrance would be that, to increase the biopsy accuracy, automatic delineation of the suspect lesion would be of greater help, especially for nodules with a smaller volume and with an intermediate PI-RADS score. Although it has the potential of aiding the radiology assistant that performs the segmentation, the urologist that further executes the annotations and overlaps the MRI acquisitions with the real-time transrectal ultrasound does not benefit equally.

CAD detection alone registered a significantly lower performance compared to radiologist’s interpretation (35.8% versus 68.64%), representing a great advantage only when used in combination with the expert’s reading. The authors attributed these findings to the CAD system’s overestimation of csPCa on DCE acquisitions and its susceptibility of being influenced by technical artifacts [44]. Additionally, it was showed that for the peripheral zone, CAD-aided biopsy did not outperform the standard procedure, significant improvement being noted only for lesions situated in the transitional and anterior part of the gland, due to their inherent heterogenous nature and great potential of false-positive results [45].

### 4.4. Limitations of the Review Process

The review process employed for the current paper has limitations as well. Being a relatively newly developed domain, we could not extend the exclusion criteria, leading to an accentuate heterogeneity amongst selected papers in terms of study protocol (MRI magnetic field intensity, selected MRI sequences, targeting all identified lesions versus only index nodules and the choice of performing external validation), algorithm complexity (automated detection and characterization of the suspect lesions versus manual segmentation and texture analysis-based malignancy prediction) and ground truth reference (prostate biopsy cores versus radical prostatectomy specimen).

### 4.5. Implications for Clinical Practice and Future Research

Although the cited articles face the above-mentioned limitations, radiomics is a rapidly developing branch of radiology, with promising results when used in combination with clinical and biochemical information [13]. Amongst the most studied pathologies depicted through MRI protocols are the central nervous system tumors, neuroradiology comprising 19.71% of the international literature tackling radiomics-based tools, being the fastest developing research domain, with an annual growth rate of published papers of 316.02%, between 2013–2018 [51]. The greatest advantage that texture analysis brings in this context is the preoperative aggressiveness assessment of gliomas, differentiating low-grade gliomas from glioblastoma multiforme with an accuracy of 89% [52], having the potential of predicting the short (<12 months) or long-term survival rate (>24 months), based on the identified heterogeneity degree [53]. Finally, this emerging research area finds applicability in benign pathologies as well, being used for instance in differentiating endometriomas from hemorrhagic ovarian cysts, outperforming the classical pathognomonic “T2 dark spots” sign in terms of sensitivity (55.17%, versus 75%) and both “T2 shading” and “T2 dark spots” signs when it comes to specificity (35.71% and 64.29% versus 100%) [54].

Even though the diagnosis and screening of PCa has been the subject of great innovations in the past decades, it is still affected by interobserver subjectivity, especially in early-detected cases that fall in the “gray area” of PI-RADS 3 lesions and PSA values between 4 and 10 ng/mL. Having texture analysis tools and computer-assisted diagnosis as additional elements that can differentiate between active-surveillance or radical treatment strategies and that can weight in upon the predicted malignancy of a suspect lesion opens the path for personalized diagnosis, offering patients tailored treatment and follow-up options.

## 5. Conclusions

The proposed systematic review shows that texture analysis and artificial intelligence can be taken into consideration when aiming to improve diagnostic precision, reaching an accuracy as high as 95% in training environment. Although the recently developed AI algorithms need further testing on large, multicentric cohorts, these techniques have the potential to serve as decision support tools that enhance the expert’s performance and bring additional information to correctly stratify the risk of each patient, opening a new horizon in terms of personalized medicine.

## Figures and Tables

**Figure 1 jpm-12-00983-f001:**
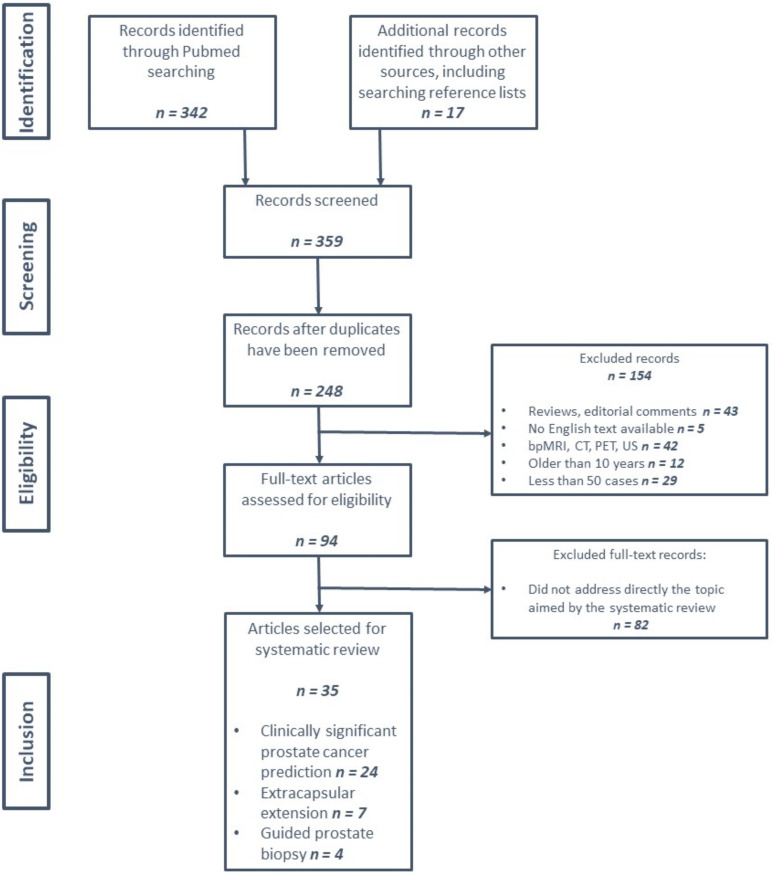
PRISMA flowchart of the screened and included studies.

**Table 1 jpm-12-00983-t001:** Features of individual studies describing strategies of improving PCa detection.

No.	Study	No. of Centers	Total Cases	Study Protocol	mpMRI Field Power (T)	Sequences Used for Features Extraction	Segmentation	Ground Truth	Focus Region
1.	Zhang et al., 2021 [10]	Unicentric	139	Training *n* = 93 Testing *n* = 46	3	T2WI DWI	Manual	Systematic prostate biopsy	PZ
2.	Bonekamp et al., 2018 [11]	Unicentric	316	Training *n* = 183 Testing *n* = 133	3	T2WI ADC	Manual	Targeted prostate biopsy	PZ + TZ
3.	Hectors et al., 2021 [12]	Unicentric	240	Training *n* = 188 Testing *n* = 52	3	T2WI	Manual	Targeted prostate biopsy	PZ + TZ (Same protocol)
4.	Zhang et al., 2021 [13]	Unicentric	140	Training *n* = 105 Testing *n* = 35	3	T2WI ADC DCE	Manual	Systematic prostate biopsy Radical prostatectomy specimen	WG
5.	Giannini et al., 2021 [14]	Multicentric	149	Training *n* = 81 Testing *n* = 38 Validation *n* = 30	1.5	T2WI ADC	Automated	Radical prostatectomy specimen	PZ
6.	Parra et al., 2019 [15]	Unicentric	72	Single cohort	1.5/3	DCE	Manual	Systematic prostate biopsy	PZ + TZ
7.	Winkel et al., 2020 [16]	Unicentric	402	Benign *n* = 201 Low risk *n* = 57 Intermediate risk *n* = 97 High risk *n* = 47	3	DCE	Manual	Targeted prostate biopsy	PZ
8.	Han et al., 2021 [17]	Unicentric	176	Training *n* = 123 Testing *n* = 53	3	ADC	Automated versus Manual	Radical prostatectomy specimen	WG
9.	Li et al., 2021 [18]	Unicentric	203	Training *n* = 141 Testing *n* = 62	3	T2WI ADC DWI DCE	Manual	Systematic prostate biopsy Radical prostatectomy specimen	PZ + TZ
10.	Zhang et al., 2021 [19]	Unicentric	316	Training *n* = 183 Testing *n* = 133	3	ADC	Manual	Targeted prostate biopsy	PZ
11.	Wang et al., 2017 [20]	Unicentric	54	Single cohort	3	T2WI DWI	Manual	Radical prostatectomy specimen	PZ + TZ
12.	Hou et al., 2020 [21]	Unicentric	263	Single cohort	3	T2WI ADC DWI	Manual	Systematic prostate biopsy Radical prostatectomy specimen	PZ + TZ (Same protocol)
13.	Castillo et al., 2021 [22]	Multicentric	204	Training *n* = 48 Testing *n* = 84 Validation *n* = 72	1.5/3	T2WI ADC DWI	Manual	Radical prostatectomy specimen	PZ + TZ
14.	Khosravi et al., 2021 [23]	Multicentric	400	Training *n* = 95 Testing *n* = 305	1.5/3	T2WI	Automated	Targeted prostate biopsy Radical prostatectomy specimen	PZ
15.	Chen et al., 2019 [24]	Unicentric	381	Benign *n* = 266 Malignant *n* = 115	3	T2WI ADC	Manual	Systematic prostate biopsy	PZ + TZ (Same protocol)
16.	He et al., 2021 [25]	Unicentric	58	Single cohort	1.5	T2WI ADC	Manual	Systematic prostate biopsy	PZ
17.	Cuocolo et al., 2019 [26]	Unicentric	75	Single cohort	3	T2WI ADC	Manual	Targeted prostate biopsy	PZ
18.	Damascelli et al., 2021 [27]	Unicentric	62	Single cohort	1.5	T2WI ADC	Semiautomated	Radical prostatectomy specimen	PZ + TZ (Same protocol)
19.	Min et al., 2019 [28]	Unicentric	280	Training *n* = 187 Testing *n* = 93	3	T2WI ADC DWI	Manual	Targeted prostate biopsy	PZ + TZ
20.	Xiong et al., 2020 [29]	Unicentric	85	Single cohort	1.5	T2WI ADC	Manual	Systematic prostate biopsy	PZ + TZ (Same protocol)
21.	Liu et al., 2021 [30]	Unicentric	466	Training and testing *n* = 324 Validation *n* = 142	3	T2WI ADC	Manual	Radical prostatectomy specimen	PZ + TZ + AFMS
22.	Sanford et al., 2020 [31]	Multicentric	1034	Training *n* = 727 Testing *n* = 212 Validation *n* = 95	3	T2WI ADC DWI	Manual	Targeted prostate biopsy	PZ + TZ
23.	Schleb et al., 2019 [32]	Unicentric	457	Training *n* = 369 Testing *n* = 88	3	T2WI ADC DWI	Manual	Targeted prostate biopsy	PZ + TZ
24.	Peng et al., 2021 [33]	Multicentric	252	Training *n* = 135 Testing *n* = 59 Validation *n* = 58	1.5	T2WI DCE	Manual	Targeted prostate biopsy	PZ

mpMRI = multiparametric magnetic resonance imaging; T2WI = T2 weighted images; ADC = apparent diffusion coefficient; DWI = diffusion weighted images; DCE = dynamic contrast enhancement; PZ = peripheral zone; TZ = transitional zone; AFMS = anterior fibromuscular stroma; WG = whole gland.

**Table 2 jpm-12-00983-t002:** Features of individual studies focusing on diagnosing extracapsular extension.

No.	Study	No. of Centers	Total Cases	Study Protocol	mpMRI Field Power (T)	Sequences Used for Features Extraction	Segmentation	Main Goal
1.	Ying Hou et al., 2021 [35]	Multicentric	849	Training *n* = 596 Testing *n* = 150 External validation *n* = 103	3	T2WI DWI ADC	Automated	Develop and validate an AI based tool to preoperatively assess ECE of localized PCa
2.	Cuocolo et al., 2021 [36]	Multicentric	193	Training *n* = 104 External validation 1 *n* = 43 External validation 2 *n* = 46	1.5/3 (2 vendors)	T2WI ADC	Manual	Build an ML model to detect ECE based on radiomics
3.	Bai et al., 2021 [37]	Multicentric	284	Training *n* = 158 Internal validation *n* = 68 External validation *n* = 58	3 (3 vendors)	T2WI ADC	Manual	Preoperative prediction of ECE using peritumoral radiomics
4.	He et al., 2021 [38]	Unicentric	273	Training *n* = 192 Testing *n* = 81	3	T2WI ADC	Manual	Radiomics model for predicting ECE and PSM
5.	Xu et al., 2020 [39]	Unicentric	115	Training *n* = 82 (35 ECE and 47 non-ECE) Testing *n* = 33 (14 ECE and 19 non-ECE)	3	T2WI DWI ADC DCE	Manual	Preoperative prediction of ECE using radiomics signature
6.	Ma et al., 2019 [40]	Unicentric	210	Training *n* = 143 Validation *n* = 67	3 (2 vendors)	T2WI	Manual	Preoperative prediction of ECE using radiomics signature, compared to radiologists’ interpretation
7.	Ma et al., 2019 [41]	Unicentric	119	Training *n* = 74 (148 bilateral samples) Validation *n* = 45 (90 bilateral samples)	3 (2 vendors)	T2WI	Manual	Preoperative prediction of side specific ECE status using radiomics signature

mpMRI = multiparametric magnetic resonance imaging; PCa = prostate cancer; T2WI = T2 weighted images; ADC = apparent diffusion coefficient; DWI = diffusion weighted images; DCE = dynamic contrast enhancement; ECE = extracapsular extension, AI = artificial intelligence; PSM = positive surgical margins.

**Table 3 jpm-12-00983-t003:** Characteristics of individual studies debating the use of computer-assisted diagnosis in targeted prostate biopsies.

No.	Study	No. of Centers	Total Cases	mpMRI Field Power (T)	Sequences Used for Features Extraction	Aim of the Study
1.	Soerensen et al., 2021 [42]	Multicentric	916 Training *n* = 805 Testing *n* = 111	1.5/3 (3 vendors)	T2WI	Deep-learning automatic segmentation of the prostate
2.	van de Ven et al., 2013 [43]	Multicentric	62	3	ADC	Assessing the required spatial alignment accuracy at MRI—guided biopsies
3.	Campa et al., 2018 [44]	Unicentric	63	3	T2WI DWI DCE	Defining the accuracy of targeted cores sampled using RAD, CAD and TiT prediction
4.	Ferriero et al., 2021 [45]	Multicentric	183 Fusion biopsy *n* = 94 CAD assisted *n* = 89	3	T2WI	Comparing the csPCA detection rate of CAD-assisted targeted biopsies versus stand-alone fusion biopsies

mpMRI = multiparametric magnetic resonance imaging; csPCa = clinically significant prostate cancer; T2WI = T2 weighted images; ADC = apparent diffusion coefficient; DWI = diffusion weighted images; DCE = dynamic contrast enhancement; RAD = lesions sampled based on mpMRI prediction alone; CAD = lesions sampled based on computer-assisted diagnosis prediction alone; TiT = target -in-target lesions, identified by both radiologist and CAD system.

## Data Availability

Not applicable.

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
