# Peer review of "More than Meets the Eye: Using Textural Analysis and Artificial Intelligence as Decision Support Tools in Prostate Cancer Diagnosis—A Systematic Review"

_jpm, 2022, doi:10.3390/jpm12060983_

Round 1
Reviewer 1 Report
In their manuscript, Telecan, et al. reviewed the advantages of textual analysis and artificial intelligence (AI) in improving prostate cancer diagnosis. The authors screened literatures that used the multiparametric magnetic resonance imaging (mpMRI) and decision support systems in diagnosing and staging of prostate cancer and selected 35 papers that meet their standards. Based on the analysis of these papers, they conclude that the combination of radiomics and AI can potentially increase the accuracy of diagnosis and staging. The paper is well written and minor changes are needed.
1. The title of section is 3.3 is not accurate. This section mainly described the use of computer assisted diagnosis in targeted prostate biopsies. Please revise it.
2. To prove that radiomics and AI can actually improve the accuracy of diagnosis, instead of simply repeating the findings from every single study, the authors should comprehensively summarize and compare the data from the available papers and present the differences with or without the utilization of radiomics and AI. Please present such results by graphs which can be better received by the authors.
Author Response
Reviewer 1
In their manuscript, Telecan, et al. reviewed the advantages of textual analysis and artificial intelligence (AI) in improving prostate cancer diagnosis. The authors screened literatures that used the multiparametric magnetic resonance imaging (mpMRI) and decision support systems in diagnosing and staging of prostate cancer and selected 35 papers that meet their standards. Based on the analysis of these papers, they conclude that the combination of radiomics and AI can potentially increase the accuracy of diagnosis and staging. The paper is well written and minor changes are needed.
We would like to thank to the reviewer for his/her appreciation.
- The title of section is 3.3 is not accurate. This section mainly described the use of computer assisted diagnosis in targeted prostate biopsies. Please revise it.
We thank the reviewer for the comment. We modified the title of the section accordingly. Please refer to page 11 of the manuscript, row 309.
- To prove that radiomics and AI can actually improve the accuracy of diagnosis, instead of simply repeating the findings from every single study, the authors should comprehensively summarize and compare the data from the available papers and present the differences with or without the utilization of radiomics and AI. Please present such results by graphs which can be better received by the authors.
We thank the reviewer for his/her suggestion. We detailed the current status of prostate cancer diagnosis, extracapsular extension and targeted prostate biopsies, along with their current hindrances. You can find the additional information at page 12, rows 346 – 354, page 13, rows 381 – 386 and page 13, rows 402 – 406.
Reviewer 2 Report
In this systematic review, the authors discussed the value of AI tools regarding textural analysis of MRI images in prostate cancer.
This review is thorough and the methodology are clear.
Minor comments:
The introduction need to be rewritten, several points are not clear or strongly stated without evidence
Examples:
Line 488: “this is not of diagnostic value,”
it has diagnostic value! use other terms to describe the need for biopsy despite suspicious mri
Line 57: “most departments, it is carried out by a 56 urologist without previous formal training. Therefor”
This is a strong statement and the authors should provide reference for it or delete it
Line 61: “Iemura et al.”
the description of iemura et al paper is not clear! iemura et al retrospectively evaluated patients undergoig robotic radical prostatectomy and found that periprostatic fat thickness on preoperative mri was a risk factor for upstaging. so this article is not related to the point the authors want to explain”
Author Response
Reviewer 2
In this systematic review, the authors discussed the value of AI tools regarding textural analysis of MRI images in prostate cancer.
This review is thorough and the methodology are clear.
We would like to thank to the reviewer for his/her appreciation
Minor comments:
Line 48: “this is not of diagnostic value,”
it has diagnostic value! use other terms to describe the need for biopsy despite suspicious mri
We thank the reviewer for the comment. mpMRI has the property of raising the suspicion of PCa with high PPV, however, in order to decide upon further therapy, histopathological confirmation is needed. We clarified the sentence. Please refer to page 2, rows 48 – 52.
Line 57: “most departments, it is carried out by a 56 urologist without previous formal training. Therefor”
This is a strong statement and the authors should provide reference for it or delete it
We thank the reviewer for the comment. We wanted to convey the message that, in most departments, the urologist is the one that does the annotation of the lesion and the synchronization of the two imagistic methods, without the assistance of the radiologist. Compared to the radiologist, the urologist does not have the experience and the same training, although special courses, masterclasses and workshops have been completed before performing the procedure. Please refer to page 2, row 60.
Line 61: “Iemura et al.”
the description of iemura et al paper is not clear! iemura et al retrospectively evaluated patients undergoig robotic radical prostatectomy and found that periprostatic fat thickness on preoperative mri was a risk factor for upstaging. so this article is not related to the point the authors want to explain”
We thank the reviewer for the suggestion. We made the proper changes. Please refer to page 2, rows 62 – 64.
Reviewer 3 Report
This is a systematic review of the integration of radiomics and AI in the diagnosis of prostate malignancies. Overall, it is well-written, interesting.
If any correction is to be made, it would be to correct the many AI technology-related terms throughout the text, which makes the text difficult to read for medical professionals. We would like to see the text written more concisely and clearly from the viewpoint of medical professionals, focusing mainly on how this research will contribute to medical care and how medical care will change in the future.
Author Response
Reviewer 3
This is a systematic review of the integration of radiomics and AI in the diagnosis of prostate malignancies. Overall, it is well-written, interesting.
We would like to thank to the reviewer for his/her appreciation.
If any correction is to be made, it would be to correct the many AI technology-related terms throughout the text, which makes the text difficult to read for medical professionals. We would like to see the text written more concisely and clearly from the viewpoint of medical professionals, focusing mainly on how this research will contribute to medical care and how medical care will change in the future.
We thank the reviewer for the comment. We identified the more difficult AI terms and offered additional definitions and information. We highlighted then in yellow in the revised manuscript.